# Supplementing Vitamin D in Different Patient Groups to Reduce Deficiency

**DOI:** 10.3390/nu15173725

**Published:** 2023-08-25

**Authors:** Pawel Pludowski

**Affiliations:** Department of Biochemistry, Radioimmunology and Experimental Medicine, The Children’s Memorial Health Institute, 04-730 Warsaw, Poland; pludowski@yahoo.com

**Keywords:** vitamin D deficiency, routine monitoring, general population, risk groups, cholecalciferol

## Abstract

Introduction: Studies indicate a high prevalence of vitamin D deficiency in both the general population and at-risk groups. Given the association between vitamin D deficiency and various diseases, addressing this concern becomes crucial, especially in situations where routine monitoring is challenging. Materials and methods: A systematic literature review of the current knowledge on vitamin D dosing in diverse at-risk populations and the application of the findings to a broader clinical perspective. Results: The reviewed studies revealed a high prevalence of vitamin D deficiency among patients with musculoskeletal disorders, systemic connective tissue diseases, corticosteroid use, endocrine and metabolic conditions, malabsorption syndromes, obesity, chronic kidney disease, cancer, and central nervous system diseases. Vitamin D deficiency was often more severe compared to the general population. Higher dosages of vitamin D beyond the recommended levels for the general population were shown to be effective in improving vitamin D status in these at-risk individuals. Additionally, some studies suggested a potential link between intermittent vitamin D administration and improved adherence. Conclusion: Simplified dosing could empower clinicians to address vitamin D deficiency, particularly in high-risk populations, even without routine monitoring. Further research is needed to establish the optimal dosing regimens for specific at-risk populations.

## 1. Introduction

Vitamin D as a mediator in the regulation of skeletal and calcium–phosphate metabolism plays an important role in muscle–bone interactions and health and in the prevention of nutritional rickets, osteomalacia, and osteoporosis. Expressions of the vitamin D receptors (VDRs) in human cells suggest an even more widespread, extra-skeletal effect of vitamin D on overall health [1]. The involvement of vitamin D in various organ and tissue effects stems from the presence of VDRs in every tissue and cell throughout the body, including the immune cells, skin, brain, gonads, stomach, heart, and pancreas. As a result, vitamin D deficiency can affect the function of these organs. A decrease in 25-hydroxyvitamin D concentrations, i.e., 25(OH)D—the major determinant of vitamin D status, is associated with numerous chronic diseases. In fact, low 25(OH)D concentrations (Table 1) were shown to be related or at least to coincide with the risk of cancer, malabsorption syndromes, osteoporosis, and other diseases and complications characterized by impaired bone metabolism, autoimmune diseases, allergies, endocrine diseases, etc. [2].

The high worldwide prevalence of vitamin D deficiency requires actions to improve this situation [1]. General screening for vitamin D deficiency is not recommended; however, 25(OH)D testing is suggested in certain risk groups prone to vitamin D deficiency, in order to find the optimal vitamin D dosing regimen to assure vitamin D sufficiency. Assessment of the 25(OH)D value is proper for overweight or obese people and for patients with chronic treatment with medications that influence the vitamin D metabolism (e.g., antiseizure medications, glucocorticoids), malabsorption syndromes (e.g., cystic fibrosis, inflammatory bowel diseases, bariatric surgery, radiation enteritis), hepatic failure, chronic kidney disease, osteomalacia, chronic musculoskeletal pain, hyperparathyroidism, autoimmune diseases (e.g., multiple sclerosis, rheumatoid arthritis), older adults (>65 years), especially those with a history of falls or nontraumatic fractures (osteoporosis), patients with granuloma-forming disorders (e.g., sarcoidosis, tuberculosis), people with chronic infections, and those with dark skin pigmentation [1].

As already mentioned, vitamin D deficiency is a worldwide problem. The primary objective of this article was, therefore, to conduct a comprehensive literature review that contains the most up-to-date recommendations for vitamin D dosing. This review article focused on providing guidance for vitamin D3 supplementation in patients of selected groups at risk of vitamin D deficiency, including those where regular monitoring of 25(OH)D is not feasible. The objective was to reduce the prevalence of vitamin D deficiency in those groups by offering suggestions for intermittent vitamin D dosages, mainly 7000 IU and 30,000 IU, which align with the number of days of the week and month. This article aimed to contribute to the overall efforts to combat vitamin D deficiency and promote optimal well-being.

## 2. Methods

A systematic literature review of vitamin D dosing was performed for the risk groups of vitamin D deficiency as well as for the general population with the use of the PubMed electronic database. The research strategy involved a combination of terms such as “vitamin D dosing”, “vitamin D supplementation”, “vitamin D deficiency”, and “risk groups”. Additional keywords were included to capture specific risk groups of interest, such as “osteoporosis”, “malabsorption”, “musculoskeletal pain” and similar. Articles published between 2007 and 2023 were included. Data extraction and thematic analyses were performed to identify common themes and trends related to vitamin D dosing practices. The analysis aimed to generate practical suggestions for vitamin D dosing in at-risk groups without routine monitoring, taking into consideration current evidence and trends identified in the literature. The final suggestions provided in this review are based on the synthesis of the findings and aim to address the specific needs of these groups of patients as well as those of the general population.

Ethical considerations were not applicable as this study relied on the analysis of the published literature.

### 2.1. Vitamin D Supplementation in the General Population

#### Intermittent Dosing 

Due to the lipophilic nature of vitamin D, weekly and monthly administrations of daily equivalent of 1000 IU of vitamin D3 provide equal efficacy and safety profiles [4]. Due to vitamin D’s 2-month half-life [5], its daily dosing is not necessary, while intermittent supplementation (e.g., weekly, monthly dosing) might be preferred by some patients. 

A randomized clinical trial comparing the efficacy of daily, weekly, and monthly administration of the same cumulative dose of vitamin D3 (equivalents of 1500 IU daily) showed the mean concentration values of 25(OH)D over a 2-month period to be similar [6].

Another study comparing different vitamin D dosing schedules (1000 IU daily, 7000 IU weekly, 30,000 IU monthly) showed equal efficacy and safety for adults with low 25(OH) D < 20 ng/mL. Increases in 25(OH)D concentration values appeared similar among the groups [7].

A Central and Eastern European Expert Consensus Statement highlights the importance of avoiding regular intake of exceedingly high vitamin D doses, such as 50,000 IU [3]. If such high doses are avoided, this statement suggests that intermittent dosing on a weekly or monthly basis can be considered, as it has the potential to improve adherence [3]. For example, in a study by Rothen et al., adherence was significantly higher with monthly rather than weekly administration of vitamin D. Moreover, monthly oral vitamin D was preferred over more frequent (weekly) administration by participants [8]. To facilitate simplicity and possibly improve adherence, recommended vitamin D daily doses for the general population [1] were recalculated based on equivalent weekly or monthly doses (Table 2).

### 2.2. Supplementation of Vitamin D3 with Dosages of 7000 IU and 30,000 IU

In 2015, a study comparing oral vitamin D3 in the doses of 7000 IU/day and 4000 IU/day in a pediatric population showed the median serum 25(OH)D concentrations were higher in the group taking 7000 IU/day at both 6 and 12 weeks. Furthermore, at 12 weeks the median serum parathyroid hormone (PTH) concentration was lower in the group exposed to 7000 IU/day (a high-dose group), and the reductions from baseline after 12 weeks were significant for depressive symptoms, fatigue, and pain [9]. 

A significant increase in mean 25(OH)D concentration from 14 ng/mL to 44 ng/mL (35 to 110 nmol/L; *p* < 0.00001) and a significant decrease in PTH (*p* < 0.05) were observed in a randomized controlled study comparing the efficacy of 7000 IU of cholecalciferol daily vs. placebo for 26 weeks. With bone mineral density (BMD) that significantly increased at the forearm by 1.6 ± 0.7% (*p* = 0.03), this study determined a positive effect of 7000 IU/day to bone turnover and BMD in obese subjects [10]. Daily supplementation with 7000 IU was efficacious in improving vitamin D status and decreasing serum PTH concentration values [9,10].

In randomized controlled clinical trial (RCT), the weekly administration of 30,000 IU of vitamin D for 12 weeks did not raise safety concerns but provided an effective tool for the normalization of 25(OH)D concentrations to the desirable value of above 30 ng/mL in deficient patients. Moreover, the limit of 25 ng/mL 25(OH)D was achieved by 95% of patients in 8 weeks with the use of 30,000 IU/week compared to only 33% using 1000 IU/day. The difference was even more prominent between dosage regiments when the 25(OH)D concentration of >30 ng/mL was analyzed: 95% vs. 24% by the end of 12 weeks of treatment, respectively [11]. 

Another study showed that the weekly administration with a dose of 30,000 IU provided vitamin D sufficiency (>30 ng/mL) in more patients with deficiency compared to a daily dose of 1000 IU [4], whereas administration of 30,000 IU twice a week for 5 weeks was shown as a rapid, effective, and safe treatment for vitamin D deficiency [4].

## 3. Vitamin D Deficiency and Supplementation with Higher Dosages in Risk Groups of Patients

### 3.1. Risk Groups with Higher Vitamin D Deficiency and Benefits of Higher Vitamin D Supplementation 

#### 3.1.1. Musculoskeletal Disorders, Such as Osteoporosis and Osteopenia

It is well-known that vitamin D is involved in bone growth and bone remodeling by osteoblasts and osteoclasts; thus, its deficiency (25(OH)D < 20 ng/mL) accelerates bone turnover, bone loss, and osteoporotic fractures [2]. Shahnazari et al. examined the vitamin D status of patients with osteoporosis and found a higher prevalence of deficiency compared to the general population [12]. 

The most recent RCT, called the “VIDA study”, where vitamin D3 was given once a month, with 200,000 IU administered to patients in the first month and then 100,000 IU administrated in the following months for 3 years, revealed that correcting severe vitamin D deficiency with 100,000 IU of vitamin D3 monthly improved BMD compared to placebo, whereas vitamin D supplementation did not have a significant impact on BMD or bone quality in already-vitamin-D-replete adults [13]. Furthermore, the hazard ratio (HR) of the first kidney stone event in these patients was 0.90 (95% CI: 0.66–1.23; *p* = 0.51) compared with the placebo arm, and the HR for the first hospitalization related to a urolithiasis event was 0.62, which was not significant (95% CI: 0.24, 1.26; *p* = 0.30). There were no cases showing hypercalcemia in the studied group; moreover, a median of 3.3 years of this study for 100,000 IU/month did not affect the incidence rate of kidney stone events nor hypercalcemia [14].

#### 3.1.2. Systemic Connective Tissue Diseases, Such as Rheumatoid Arthritis, Fibromyalgia, and Chronic Musculoskeletal Pain

A systematic review of 81 studies found that people suffering from arthritis and pain tended to have decreased 25(OH)D concentrations compared to healthy individuals [15].

Lombardo et al. reviewed the efficacy of vitamin D supply in the treatment of chronic musculoskeletal pain (CMP) and fibromyalgia syndrome (FMS), suggesting that low 25(OH)D values, reflecting vitamin D deficiency, may be associated with an increased risk of FMS and CMP [12].

A large RCT study (VITAL study) involving 25,871 participants found that the group receiving 2000 IU of cholecalciferol daily had a 22% lower risk of developing new autoimmune diseases compared to the placebo group. This effect became even stronger when the first two years of follow-up were excluded, confirming the protective role of vitamin D supplementation in reducing the incidence of autoimmune diseases [16].

#### 3.1.3. Glucocorticoid-Induced Osteoporosis

Drug-induced osteoporosis is a common form of secondary osteoporosis, with glucocorticoids being a crucial component of immune suppression and anti-inflammatory treatments for conditions like arthritis [17]. It is estimated that around 40% of patients undergoing long-term glucocorticoid therapy experience fractures at some point during their lifetime. Initially, glucocorticoids increase bone resorption and decrease bone formation. Their long-term usage primarily suppresses bone formation. Glucocorticoids also affect calcium homeostasis, parathyroid gland function, and the vitamin D metabolism, indirectly impacting bones. Additionally, they contribute to muscle loss, increasing the risk of falls and fractures [17]. 

The association of steroid use with 25(OH)D deficiency was shown, with a statistically significant greater percentage of steroid users having 25(OH)D concentrations lower than 10 ng/mL compared to those who did not use steroids [18].

As chronic exposure to glucocorticoid excess can affect the vitamin D metabolism, vitamin D supplementation was found to be even more important in such cases. Resistance to vitamin D dictates a higher level of 25(OH)D to achieve ≥32 ng/mL; however, the daily administration of 2000 IU of vitamin D is often sufficient to reach and maintain those optimal concentrations [19].

#### 3.1.4. Endocrine and Metabolic Conditions, Such as Diabetes Mellitus (Type 1 and 2), Metabolic Syndrome, Hypo- and Hyperparathyroidism, etc. 

Calcitriol, the active form of vitamin D, was found to influence the β-cells in the pancreas, in addition to exerting some effects on insulin secretion. One of the reasons behind the association of vitamin D deficiency with insulin resistance is presence of vitamin D receptors (VDRs) and the enzyme 1-α hydroxylase, crucial for calcitriol synthesis, within pancreatic β-cells. Multiple studies had documented a possible link between vitamin D and pancreatic β-cells function in which vitamin D deficiency can lead to prediabetes, and even diabetes itself [20].

The presence of vitamin D receptors and the enzyme 1-α hydroxylase, crucial for calcitriol synthesis, within pancreatic β-cells was documented, and vitamin D deficiency was shown to change pancreatic β-cells function leading to prediabetes and even diabetes itself [20].

In a cross-sectional study conducted by Utmani et al. involving 174 patients, it was observed that individuals with a metabolic syndrome had significantly lower mean serum 25(OH)D concentration values compared to those without the syndrome [21]. Studies concluded that low vitamin D supply was associated with decreased insulin sensitivity, increased insulin resistance (IR), and high fasting blood glucose (FBG) [20].

Findings in a meta-analysis conducted by Taheriniya et al. revealed a significant correlation between lower 25(OH)D concentration values and autoimmune thyroid diseases, Hashimoto’s thyroiditis, and hypothyroidism. It was concluded that vitamin D deficiency is highly prevalent in endocrine disorders, and its supplementation might have numerous beneficial effects [22].

In a large RCT involving patients with prediabetes at a high risk of progressing to type 2 diabetes (T2D), supplementation with 4000 IU/day of vitamin D showed a tendency, although not statistically significant, toward a slower progression to T2D compared to placebo. However, a post hoc analysis of patients without obesity, severe vitamin D deficiency at baseline, and excellent adherence to vitamin D treatment revealed a significant effect in reducing the progression to T2D [23]. 

A meta-analysis of studies involving 1722 women found that vitamin D supplementation exceeding 2000 IU/day reduced the incidence of gestational diabetes mellitus (GDM) compared to lower doses (≤2000 IU/day) [24].

Furthermore, an RCT examined the effects of vitamin D supplementation on inflammatory markers in non-obese patients with type 2 diabetes. The study demonstrated that 30,000 IU of cholecalciferol per week for six months led to higher 25(OH)D concentrations and a decrease in certain inflammatory markers compared to the placebo group [25].

In patients with endocrine conditions, such as primary hyperparathyroidism, studies demonstrated that vitamin D supplementation of up to 2800 IU per day is safe and is associated with reductions in PTH without affecting calcium or creatinine concentration values [26].

#### 3.1.5. Obesity 

Many potential pathophysiological mechanisms were considered to explain the relation between vitamin D, its deficiency, and the state of obesity; the problem is widely recognized and supported by medical data. Human body fat as a storage for vitamin D, if excessive, may alter the kinetics between the depot and circulation. Furthermore, obese people are more likely to limit their outdoor physical activity, to avoid exposure of their bodies to sunlight, are prone to consume a diet with low amounts of vitamin D, and reveal alterations in VDRs and impaired hydroxylation in adipose tissue [19]. It is considered that overweight and obese subjects are more resistant to vitamin D supplementation compared to lean individuals, as several studies indicated that obese subjects have lower serum 25(OH)D concentration values by approximately 15.2 ng/mL (38 nmol/L), and obese children have a 45% reduction in circulating values after the administration of equal doses of vitamin D [19].

A review by Bleizgys (2021) recommended, for risk-group patients with vitamin D deficiency, particularly obese individuals and persons with malabsorption syndromes, an increase in their vitamin D dose by two-fold or sometimes even three-fold, pointing out that vitamin D doses of up to 10,000 IU daily are considered safe for the vast majority of patients [27]. The same was proposed in the guidelines for Poland in 2018 and 2023 [1,28].

#### 3.1.6. Malabsorption Syndromes, Such as Inflammatory Bowel Disease, Crohn’s Disease, Cystic Fibrosis, Ulcerative Colitis, Celiac Disease, etc.

Due to its fat-soluble nature, vitamin D requires bile salts for its absorption, which occurs mainly in the duodenum, by successively forming micelles and chylomicrons for its transport [29]. Consequently, individuals with malabsorption disorders like inflammatory bowel diseases, pancreatic insufficiency, celiac disease, cystic fibrosis, cholestatic liver diseases, and short bowel syndrome are more prone to vitamin D deficiency [2,30]. 

In chronic inflammatory conditions affecting the gastrointestinal tract, such as Crohn’s disease, ulcerative colitis, and celiac disease, vitamin D was found to play a role in reducing inflammation and maintaining gut microbiota. Patients with these conditions are more likely to experience vitamin D deficiency compared to the general population, primarily due to malabsorption issues. Their vitamin D deficiency can contribute to the development of osteopenia and osteoporosis [2]. 

An RCT demonstrated that short-term treatment with 2000 IU/day of vitamin D increased 25(OH)D concentrations in Crohn’s disease cases, enhanced antimicrobial peptide LL-37 concentrations, and maintained intestinal permeability [30].

The combination of calcium and vitamin D supplementation is suggested to be specifically targeted toward individuals at risk of osteoporosis with coinciding intestinal malabsorption or following bariatric surgery. However, routine intermittent doses of ≥60,000 IU are not recommended due to an increased risk of fracture and falls [28,31].

#### 3.1.7. Chronic Kidney Disease (CKD)

Vitamin D deficiency is prevalent and even worsens during CKD and its progression. A study by Caravaca-Fontán et al. demonstrated that more than 80% of 367 pre-dialysis patients had 25(OH)D concentrations <20 ng/mL, and their reduced kidney function affected both the anabolic and catabolic phases of the vitamin D metabolism [32].

Oksa et al. conducted an RCT involving 87 patients with CKD stages 2–4, comparing low-dose (5000 IU/week) and high-dose (20,000 IU/week) cholecalciferol supplementation [33]. After 12 months, both groups showed a significant increase in plasma 25(OH)D concentrations, with the high-dose group exhibiting a statistically higher increase compared to the low-dose group. Plasma PTH concentrations significantly decreased in both groups without significant differences between them, indicating that high-dose cholecalciferol was more effective in increasing 25(OH)D concentrations [33].

In France, nephrologists commonly prescribed monthly oral doses of 100,000 IU cholecalciferol, achieving normalization of serum 25(OH)D concentration values in over 85% of cases [34]. In Belgium, Delanaye et al. reported the successful attainment of the recommended targets (>30 ng/mL) after 12 months of using oral cholecalciferol 25,000 IU every two weeks [35].

#### 3.1.8. Cancer

Vitamin D deficiency is highly prevalent among cancer patients, with one study revealing that 72% of individuals diagnosed with cancer had an insufficient vitamin D status [36]. Several systematic reviews and meta-analyses examining the correlation between 25(OH)D concentration values and mortality outcomes in cancer patients showed a protective effect of higher concentrations in various types of cancers, including breast cancer, colorectal cancer, prostate cancer, and hematological malignancies [37]. Moreover, some studies suggested that initial vitamin D deficiency (<20 ng/mL) and insufficient repletion might be associated with a worse prognosis in patients with metastatic melanoma [38]. 

The results of meta-analyses confirmed the protective effect of vitamin D supplementation or the optimal vitamin D status on cancer mortality but showed no significant effect on cancer incidence [39].

The VITAL study, uniquely focused on cancer mortality, showed a significant difference of approximately 12 ng/mL (30 nmol/L) in 25(OH)D concentration between the intervention, receiving a daily dose of 2000 IU of vitamin D, and the placebo-controlled group. Notably, the study demonstrated increasing benefits over time, as the relative mortality risk decreased to 0.75 (95% CI, 0.59–0.96) when the first two years of observation during this study were omitted [40].

Efforts to achieve 25(OH)D values around 21–54 ng/mL (54–135 nmol/L) may contribute to reducing cancer mortality and to consistently raising the concentration above 30 ng/mL (75 nmol/L), meaning at least 1500–2000 IU/day is required for adults [41]. Intermittent dosing in severely ill patients taking multiple medications daily is currently still limited. However, based on all the reviewed articles demonstrating advantages in several risk groups, it holds promise as a potentially advantageous approach for them. 

#### 3.1.9. Immunocompromisation, e.g., Caused by HIV Infection 

Observational studies reported a high prevalence of vitamin D deficiency among people living with HIV [42]. In an RCT, safety and efficacy in improving vitamin D status was determined with a daily supplementation of 7000 IU in HIV patients, monitored over 12 months. A daily supplementation of 7000 IU was efficacious in improving vitamin D status, and the safety events were not recorded [43].

Moreover, a review of several studies on the potentially protective role of vitamin D supplementation ranging from 400 to 14,000 IU of vitamin D daily on HIV-1 infection showed that the use of 7000 IU daily was the most effective dose, which restored vitamin D sufficiency (>30 ng/mL) in 80% of patients, with higher concentrations observed following 12 months of treatment [44].

#### 3.1.10. Central Nervous System Diseases, Such as Multiple Sclerosis, Epilepsy, Dementia, Alzheimer’s Disease, Parkinson’s Disease, etc.

The evidence showed that reduced 25(OH)D concentrations are associated with an increased incidence of multiple sclerosis (MS) and its progression, and VDRs were found in neural cells, suggesting the meaning of optimal vitamin D status, through mechanistic pathways, for MS disease [19].

Geng et al. observed that lower serum 25(OH)D values are associated with an increased risk of dementia and Alzheimer’s disease [45].

A study by Jesus et al. showed that patients with epilepsy are more often obese and that vitamin D deficiency is more common, with a much higher prevalence of severe deficiency compared to the general population. Furthermore, certain anti-epileptic drugs, such as phenytoin, valproate, and carbamazepine, are known to decrease both 25(OH)D and 1,25(OH)_2_D concentration values due to the stimulation of the metabolic clearance of these metabolites to inactive forms through the catabolic pathway. Therefore, people with epilepsy face a six-fold risk of bone fracture compared to the general population, which is probably due to the interplay among frequent falls, reduced bone density, and low vitamin D status, i.e., 25(OH)D concentrations reflecting vitamin D deficiency (<20 ng/mL) or insufficiency (<30 ng/mL but >20 ng/mL) [46].

In a small, randomized controlled trial, high-dose vitamin D supplementation (10,000 IU/day for 4 months) did not appear to improve balance in Parkinson’s disease patients compared to placebo. However, a post hoc analysis revealed that younger patients (52–66 years old) experienced improved balance compared to older participants [47].

A study involving women with T2D and depressive symptoms showed that 50,000 IU of ergocalciferol given once per week for 6 months resulted in a significant decrease in depression and anxiety [48].

In a 12-month study by Suzuki et al., Parkinson’s disease patients receiving 1200 IU of vitamin D daily showed doubled serum 25(OH)D concentrations and stable motor scores, while the placebo group experienced a significant decline in motor scores with unchanged 25(OH)D values. This suggested that vitamin D administration may help stabilize motor symptoms in Parkinson’s disease, at least in the short term [49].

To conclude, all the above-mentioned studies highlight the increased risk of vitamin D deficiency or a higher need for vitamin D in various at-risk populations, including those with musculoskeletal disorders, systemic connective tissue diseases, corticosteroid use, endocrine and metabolic conditions, malabsorption syndromes, obesity, chronic kidney disease, cancer, immunocompromisation, and even central nervous system diseases. The findings emphasize the need for tailored vitamin D supplementation strategies in these specific risk groups to address and mitigate deficiencies effectively.

### 3.2. Safety of Very High Dosages 

It is important to acknowledge that the use of higher doses of vitamin D carries potential risks, albeit rare, of adverse effects related to excessive supplementation. These adverse effects primarily arise from the development of hypercalcemia and hypercalciuria. Although overt hypercalcemia is uncommon, certain studies indicated that daily vitamin D supplementation exceeding 4000 IU may potentially compromise bone health and elevate the risk of falls. Additionally, the concurrent supplementation of vitamin D and calcium was associated with an increased risk of kidney stone formation, at least without proper hydration of the organism. Consequently, it is essential to recognize that vitamin D supplementation far beyond the recommended daily allowances should not be regarded as a harmless intervention [18]. The Guidelines for Preventing and Treating Vitamin D Deficiency: A 2023 Update in Poland recommend, as an upper limit for daily cholecalciferol intake for vitamin D deficiency prophylaxis in normal-weight adults (>19 years old), 4000 IU daily and, for those who are overweight or obese, 10,000 IU daily [1]. Additionally, vitamin D intoxication usually was noted in persons who decided, without a medical doctor’s prescription, to take very large doses (e.g., 50,000–100,000 IU/day) of vitamin D for several months to several years [50,51].

## 4. Discussion

There is a high prevalence of vitamin D deficiency in both the general population [1] as well as in risk groups, such as patients with musculoskeletal disorders, systemic connective tissue diseases, corticosteroid use, endocrine and metabolic conditions, malabsorption syndromes, obesity, chronic kidney disease, cancer, immunocompromisation, and even with central nervous system diseases. Several reviewed studies showed the benefits of vitamin D supplementation for various risk groups, often administered in higher dosages compared to the recommendations for the general population. Vitamin D supplementation improves musculoskeletal health by enhancing BMD and reducing the risk of osteoporosis. Vitamin D also plays a role in preventing autoimmune diseases and pain, managing glucocorticoid-induced osteoporosis, and decreasing parathyroid hormone activity, followed by positive effects on pain. Additionally, it shows promise in preventing the progression of prediabetes to T2D, reducing the incidence of gestational diabetes as well as cancer mortality. In patients with a malabsorption syndrome, such as Crohn’s disease, vitamin D boosts the levels of antimicrobial peptides and helps maintain the integrity of the intestines. Its higher supplementation is of great importance in chronic kidney disease patients, because various factors of impaired kidney function contribute to vitamin D deficiency, affecting its production, activation, and degradation. The potential benefit of high-dose vitamin D supplementation in a central nervous system disorder is shown by improved balance in Parkinson’s disease, reduced depression and anxiety, and a positive effect on the stabilization of motor symptoms in Parkinson’s disease patients. Overall, ensuring adequate vitamin D intakes is crucial for maintaining optimal health across a wide range of conditions [2,12,13,14,15,16,17,18,19,20,21,22,23,24,25,26,27,28,29,30,31,32,33,34,35,36,37,38,39,40,41,42,43,44,45,46,47,48,49].

Although vitamin D supplementation can be beneficial, it is important to be cautious of the potential risks associated with higher doses. Excessive supplementation can lead to rare adverse effects like hypercalcemia and hypercalciuria. The concurrent use of vitamin D and calcium supplements may also raise the risk of kidney stone formation. Therefore, exceeding the recommended daily allowances of vitamin D should not be considered a harmless intervention. The recommended upper limit for daily cholecalciferol intake for vitamin D deficiency prophylaxis is 4000 IU daily for normal-weight adults and 10,000 IU daily for those who are overweight or obese [1].

The treatment of vitamin D deficiency in otherwise healthy patients (the general population) with up to 7000 IU of vitamin D per day should be sufficient to maintain year-round 25(OH)D concentration values between 40 and 70 ng/mL. In vitamin D deficient patients suffering from serious illnesses such as cancer, heart disease, multiple sclerosis, diabetes, autism, and a host of other illnesses, dosing schedules should be more aggressive than in healthy ones and be sufficient to obtain and maintain higher year-round 25(OH)D values, i.e., between 55 and 70 ng/mL. Moreover, vitamin D should always be an adjuvant treatment in patients with serious illnesses but never as a replacement for standard treatment [13].

Despite the fact that several of the above-mentioned risk groups suffer from hypovitaminosis D, there is currently no available guidance for clinicians for vitamin D3 dosing without prior testing. Guidelines recommend that 25(OH)D measurements should be considered for risk-group patients, and in cases where this is not possible the dosing recommendations for the general population should be followed [1]. Nevertheless, due to the high prevalence of vitamin D deficiency worldwide and the positive effects of higher vitamin D supplementation in risk groups, the aim was to draft simplified, easy-to-follow suggestions for clinicians when dealing with adult risk-group patients without 25(OH)D testing.

In clinical practice, in situations where there exists a well-established correlation between a patient’s medical condition or risk factors and vitamin D deficiency, healthcare professionals may consider the option of prescribing vitamin D supplementation without the need for routine monitoring of 25(OH)D concentrations. This suggested approach is driven by the understanding that the potential benefits of vitamin D supplementation in addressing the specific health condition or risk factors outweigh the necessity of regular monitoring. Simplification was proposed in light of the benefits of different (higher) dosages of vitamin D3 for the risk groups mentioned in the studies and the recommended highest daily dose of 4000 IU. For risk-group patients, vitamin D3 dosing of 2000–4000 IU/day (or up to 30,000 IU weekly or up to 120,000 IU monthly for 3 months)—which is also the guidelines’ [1] recommended dosing regimen for healthy individuals older than 75 years—could be adopted. 

By adopting this approach, healthcare professionals can focus on implementing evidence-based interventions that address the patient’s specific medical condition or risk factors, while considering the well-documented role of vitamin D in promoting health. It is important to note that individual patient factors and circumstances should be carefully evaluated, and clinical judgment should guide the decision-making process.

Furthermore, several studies demonstrated positive outcomes with both daily and intermittent dosing regimens of vitamin D, including strengths of 7000 and 30,000 IU [4,7,9,10,43,44]. These findings suggest that simplifying vitamin D dosing regimens could enhance patient adherence and reduce vitamin D deficiency. Logical strengths such as 7000 and 30,000 IU, which are aligned with the number of days of the week and month, respectively, can be simply recalculated and adopted. Therefore, the purpose of the suggested recalculation of daily doses to intermittent weekly and monthly regimens is to offer clinicians an additional tool to tailor therapy according to individual patient needs, ultimately aiming to improve treatment compliance and reduce the burden of vitamin D deficiency for better patient outcomes. For example, a recent paper by Takacs et al. provided the results of a clinical study of use of 30,000 IU given twice per week for five weeks and then every second week for the maintenance of optimal 25(OH)D values (30–50 ng/mL) in vitamin D deficient patients [4]. It appeared that, without negative effects on bone metabolism, calcium levels, or the risk of falls, the studied dosing schedule was effective in obtaining and maintaining proper vitamin D status [4]. 

Nevertheless, this review article has limitations. Firstly, the availability of high-quality studies on vitamin D supplementation may have limited our findings. Secondly, our focus on articles demonstrating positive outcomes of intermittent supplementation excluded those reviews that found no benefits or those demonstrating regular daily vitamin D dosing to be more beneficial for certain patient populations. Therefore, the analysis may not encompass all perspectives. Thirdly, the generalizability of our findings regarding adherence to intermittent vitamin D dosing may be limited to settings where patients have a greater sense of safety with daily dosing or where higher doses for intermittent regimens are not accessible. In conclusion, suggestions for vitamin D dosing for both the general population and at-risk groups remain below the threshold of 4000 IU cholecalciferol daily (or equivalent weekly/monthly dose), considering the potential risks associated with higher dosages. By adhering to these suggestions, this article aims to provide patients with the benefits of vitamin D supplementation while maximizing their adherence and minimizing the potential adverse effects associated with excessive doses. Nonetheless, it is crucial to emphasize that the appropriateness of foregoing routine monitoring should be determined on a case-by-case basis, considering the patient’s unique characteristics, clinical presentation, and professional judgment. Continued research and the development of clinical guidelines will contribute to refining the approach to monitoring and supplementation strategies in various populations and health conditions.

## 5. Conclusions

The high prevalence of vitamin D deficiency underscores the critical need to proactively address this issue. Even though the current guidelines [1] provide recommendations for vitamin D according to the routine monitoring of 25(OH)D concentration values, this paper provides clinicians with simple suggestions on vitamin D dosing even in the absence of routine monitoring. This approach becomes particularly useful for at-risk groups that are even more susceptible to vitamin D insufficiency. With clear and simple vitamin D dosing suggestions (aligned with the number of days of the week and month), we can empower clinicians to contribute to the overall reduction in vitamin D deficiency rates and consequently strive toward improved public health outcomes. Nevertheless, more research is necessary to determine optimal vitamin D dosing regimens for certain at-risk groups suffering from vitamin D deficiency.

## Figures and Tables

**Table 1 nutrients-15-03725-t001:** Target 25(OH)D concentration thresholds [3].

25(OH)D Concentration	Vitamin D Status
<20 ng/mL (<50 nmol/L)	Vitamin D deficiency
20–30 ng/mL (50–75 nmol/L)	Vitamin D insufficiency
30–50 ng/mL (75–125 nmol/L)	Vitamin D sufficiency
50–60 ng/mL (125–150 nmol/L)	Safe but not a target concentration
60–100 ng/mL (150–250 nmol/L)	Area of uncertainty with potential benefits or risks
>100 ng/mL (>250 nmol/L)	Potential vitamin D toxicity (oversupply)

**Table 2 nutrients-15-03725-t002:** Vitamin D3 dosing recommendations for the general population [1] with recalculated weekly and monthly dosages.

Patient Age	Risk Factor for Vitamin D Deficiency	Recommended Dosing Regimen for Preventing Vitamin D Deficiency (IU)
<65 (18+) yrs.	Insufficient **sun** exposure (from May–Sept, between 10 am and 3 pm)	1000–2000/day OR **7000–14,000/week** OR **30,000–60,000/month**
65–75 yrs.	Decreased **efficacy of skin’s synthesis** of vitamin D	
75+ yrs.	Decreased **efficacy of skin’s synthesis** of vitamin D; potential **malabsorption** and **altered metabolism**	2000–4000/day OR **14,000–30,000/week**

## Data Availability

Not applicable.

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
