# Peer review of "Supplementing Vitamin D in Different Patient Groups to Reduce Deficiency"

_nutrients, 2023, doi:10.3390/nu15173725_

Round 1
Reviewer 1 Report
Summary: The manuscript is a literature survey of vitamin D supplementation with the aim of providing guidance to clinicians. Additionally, vitamin D levels and their importances in a variety of specific disease conditions was reviewed. Author offers guidance for supplementation on daily/weekly/monthly basis [which ever frequency would enhance compliance] even in the absence of routine 25(OH)D monitoring. Also, there are citations of works that show adverse reactions are rare in patients given levels of supplementation that would be considered high/excessive by some authorities.
It is possible readers from some locations in the world may find the guidance offered here "too aggressive." Thus, author might want to comment briefly on what constitutes optimal vitamin D levels as that remains a topic of discussion in some nations. Here in the USA, the 2011 National Academy report offered guidance https://www.ncbi.nlm.nih.gov/books/NBK56070/. However, some organizations believed their recommendations were too low; in particular, the Endocrine Society https://pubmed.ncbi.nlm.nih.gov/21646368/. The UK NHS continues to be quite conservative on supplementation. How much variation in recommendations are there among various nations?
Author might want to comment briefly on the value of supplementation with 25(OH)D3 instead of vitamin D3. Would there be some circumstances where this approach could be taken and if so, what guidance could be offered?
Author Response
Reviewer 1.
Summary: The manuscript is a literature survey of vitamin D supplementation with the aim of providing guidance to clinicians. Additionally, vitamin D levels and their importances in a variety of specific disease conditions was reviewed. Author offers guidance for supplementation on daily/weekly/monthly basis [which ever frequency would enhance compliance] even in the absence of routine 25(OH)D monitoring. Also, there are citations of works that show adverse reactions are rare in patients given levels of supplementation that would be considered high/excessive by some authorities.
It is possible readers from some locations in the world may find the guidance offered here "too aggressive." Thus, author might want to comment briefly on what constitutes optimal vitamin D levels as that remains a topic of discussion in some nations. Here in the USA, the 2011 National Academy report offered guidance https://www.ncbi.nlm.nih.gov/books/NBK56070/. However, some organizations believed their recommendations were too low; in particular, the Endocrine Society https://pubmed.ncbi.nlm.nih.gov/21646368/. The UK NHS continues to be quite conservative on supplementation. How much variation in recommendations are there among various nations?
Author’s response: I want to thank Reviewer 1 for this comment. Indeed, the question is still opened for discussions not only in US but in the whole World. The answer was already given by my colleagues and by me in several papers published so far. The answer depends on the major topic of specialty of medical doctors who meet in their meetings, or in other words: please consider difference between “bone-centric guidelines” (focusing on calcium-phospate and skeleton issues) and “pleiotropic-centric guidelines” (focused on whole metabolism and diseases). But the main aim was to provide clinicians a review on daily/weekly/monthly schedules with the use of vitamin D.
Reviewer 1: Author might want to comment briefly on the value of supplementation with 25(OH)D3 instead of vitamin D3. Would there be some circumstances where this approach could be taken and if so, what guidance could be offered?
Author’s response: I want to thank Reviewer 1 also for this comment. The calcifediol is extremely important, as Reviewer 1 notes, and both the guidelines and recommended dosing were published in : Nutrients. 2023 Jan 30;15(3):695. doi: 10.3390/nu15030695. Thank you for this question and for time spent on reading and reviewing my paper.
Reviewer 2 Report
The manuscript "Supplementing Vitamin D in Different Patient Groups to Reduce Deficiency" submitted for review in Nutrients is an excellent review.
The author considers the high prevalence of 25-hydroxyvitamin D (25OH D) deficiency (a surrogate for the nutritional status of the vitamin D endocrine system) in the general population, which is even higher in several at-risk populations: patients with musculoskeletal disorders, systemic connective tissue diseases, corticosteroid use, endocrine and metabolic conditions, malabsorption syndromes, obesity, chronic kidney disease, cancer and diseases of the central nervous system.
Based on the relationship between 25OHD deficiency and these processes, and the difficulty of carrying out substitution treatment in routine clinical practice, the author reviews different substitution regimens and therapeutic regimens in the general population and the use of higher doses in different risk groups, assessing their efficacy and safety with the available data.
The author proposes the application of the results from a practical clinical perspective with the aim of making it easier for interested clinicians to understand and manage the problem of substitution treatment using native vitamin D without the need for routine 25OHD quantification.
Although it is commonly used by most authors in their scientific publications to refer to the quantification of 25OHD as vitamin D levels, this fact constitutes a serious conceptual error, which we should all eradicate, so I formally propose to the author that when he refers to measured or measurable levels of 25OHD, he should refer to them as 25OHD levels and not as vitamin D levels. That what is referred to as vitamin D deficiency should be changed to 25OHD deficiency and that this should be changed throughout the text.
On the other hand, vitamin D is metabolically inert, It should not cite the so-called actions of vitamin D as such throughout the manuscript, but rather refer to them as the actions of the vitamin D endocrine system or calcitriol.
Although the author's approach seems to me ingenious and laborious to bring a common problem closer to medical practice, he should indicate more explicitly the convenience of controlling universally accessible parameters such as calciuria corrected by creatinine in 24-hour urine and calcaemia to regulate treatment and if they are elevated, consider the determination of 25OHD
Author Response
Reviewer 2.
The manuscript "Supplementing Vitamin D in Different Patient Groups to Reduce Deficiency" submitted for review in Nutrients is an excellent review.
Author’s response: I want to thank Reviewer 2 for such nice comment.
The author considers the high prevalence of 25-hydroxyvitamin D (25OH D) deficiency (a surrogate for the nutritional status of the vitamin D endocrine system) in the general population, which is even higher in several at-risk populations: patients with musculoskeletal disorders, systemic connective tissue diseases, corticosteroid use, endocrine and metabolic conditions, malabsorption syndromes, obesity, chronic kidney disease, cancer and diseases of the central nervous system.
Based on the relationship between 25OHD deficiency and these processes, and the difficulty of carrying out substitution treatment in routine clinical practice, the author reviews different substitution regimens and therapeutic regimens in the general population and the use of higher doses in different risk groups, assessing their efficacy and safety with the available data. The author proposes the application of the results from a practical clinical perspective with the aim of making it easier for interested clinicians to understand and manage the problem of substitution treatment using native vitamin D without the need for routine 25OHD quantification.
Author’s response: I want to thank Reviewer 2 for these comments.
Reviewer 2: Although it is commonly used by most authors in their scientific publications to refer to the quantification of 25OHD as vitamin D levels, this fact constitutes a serious conceptual error, which we should all eradicate, so I formally propose to the author that when he refers to measured or measurable levels of 25OHD, he should refer to them as 25OHD levels and not as vitamin D levels. That what is referred to as vitamin D deficiency should be changed to 25OHD deficiency and that this should be changed throughout the text. On the other hand, vitamin D is metabolically inert, It should not cite the so-called actions of vitamin D as such throughout the manuscript, but rather refer to them as the actions of the vitamin D endocrine system or calcitriol.
Author’s response: I want to thank Reviewer 2 for above comment. You are totally correct. I agree with you that vitamin D (cholecalciferol and/or ergocalciferol; depending on point of discussion) seems metabolically inert (at least basing on the available scince) compared to calcitriol and vitamin D levels are not the same to calcifediol levels, and the “levels” are not the same as “serum concentrations” that is why I tried to avoid above the use of such “words”, keeping in mind the major scope you have highlighted.
Reviewer 2: Although the author's approach seems to me ingenious and laborious to bring a common problem closer to medical practice, he should indicate more explicitly the convenience of controlling universally accessible parameters such as calciuria corrected by creatinine in 24-hour urine and calcaemia to regulate treatment and if they are elevated, consider the determination of 25OHD.
Author’s response: I want to thank Reviewer 2 for comment. Again you are right. The calciuria, calcemia should be considered in a patients care but this is a case of another issue like national guidelines or previous guides:
1) Nutrients. 2023 Jan 30;15(3):695. doi: 10.3390/nu15030695. Guidelines for Preventing and Treating Vitamin D Deficiency: A 2023 Update in Poland. PaweÅ‚ PÅ‚udowski 1, Beata Kos-KudÅ‚a 2, MieczysÅ‚aw Walczak 3, Andrzej Fal 4, Dorota ZozuliÅ„ska-ZióÅ‚kiewicz 5, Piotr Sieroszewski 6, JarosÅ‚aw Peregud-Pogorzelski 7, Ryszard Lauterbach 8, Tomasz Targowski 9, Andrzej LewiÅ„ski 10, Robert SpaczyÅ„ski 11, MirosÅ‚aw WielgoÅ› 12, JarosÅ‚aw Pinkas 13, Teresa Jackowska 14, Ewa Helwich 15, Artur Mazur 16, Marek RuchaÅ‚a 17, Arkadiusz Zygmunt 10, MieczysÅ‚aw Szalecki 18, Artur Bossowski 19, Justyna Czech-Kowalska 20, Marek Wójcik 1, Beata Pyrżak 21, MichaÅ‚ A Å»mijewski 22, PaweÅ‚ Abramowicz 23, Jerzy Konstantynowicz 23, Ewa Marcinowska-Suchowierska 24, Andrius Bleizgys 25, Spirydon N Karras 26, William B Grant 27, Carsten Carlberg 28, Stefan Pilz 29, Michael F Holick 30, Waldemar Misiorowski 31